# Large-Scale Wasserstein Gradient Flows

**Petr Mokrov**[*]
Skolkovo Institute of Science and Technology
Moscow Institute of Physics and Technology
*Moscow, Russia*
petr.mokrov@skoltech.ru

**Alexander Korotin***
Skolkovo Institute of Science and Technology
Artificial Intelligence Research Institute
*Moscow, Russia*
a.korotin@skoltech.ru

**Lingxiao Li**
Massachusetts Institute of Technology
*Cambridge, Massachusetts, USA*
lingxiao@mit.edu

**Aude Genevay**
Massachusetts Institute of Technology
*Cambridge, Massachusetts, USA*
aude.genevay@gmail.com

**Justin Solomon**
Massachusetts Institute of Technology
*Cambridge, Massachusetts, USA*
jsolomon@mit.edu

**Evgeny Burnaev**
Skolkovo Institute of Science and Technology
Artificial Intelligence Research Institute
*Moscow, Russia*
e.burnaev@skoltech.ru

## Abstract

Wasserstein gradient flows provide a powerful means of understanding and solving many diffusion equations. Specifically, Fokker-Planck equations, which model the diffusion of probability measures, can be understood as gradient descent over entropy functionals in Wasserstein space. This equivalence, introduced by Jordan, Kinderlehrer and Otto, inspired the so-called JKO scheme to approximate these diffusion processes via an implicit discretization of the gradient flow in Wasserstein space. Solving the optimization problem associated to each JKO step, however, presents serious computational challenges. We introduce a scalable method to approximate Wasserstein gradient flows, targeted to machine learning applications. Our approach relies on input-convex neural networks (ICNNs) to discretize the JKO steps, which can be optimized by stochastic gradient descent. Unlike previous work, our method does not require domain discretization or particle simulation. As a result, we can sample from the measure at each time step of the diffusion and compute its probability density. We demonstrate our algorithm's performance by computing diffusions following the Fokker-Planck equation and apply it to unnormalized density sampling as well as nonlinear filtering.

## 1 Introduction

Stochastic differential equations (SDEs) are used to model the evolution of random diffusion processes across time, with applications in physics [63], finance [22, 52], and population dynamics [35]. In machine learning, diffusion processes also arise in applications filtering [34, 21] and unnormalized posterior sampling via a discretization of the Langevin diffusion [70].

The time-evolving probability density $\rho_t$ of these diffusion processes is governed by the Fokker-Planck equation. Jordan, Kinderlehrer, and Otto [32] showed that the Fokker-Planck equation is

---

[*]Equal contribution.

35th Conference on Neural Information Processing Systems (NeurIPS 2021).

equivalent to following the gradient flow of an entropy functional in Wasserstein space, i.e., the space of probability measures with finite second order moment endowed with the Wasserstein distance. This inspired a simple minimization scheme called JKO scheme, which consists an implicit Euler discretization of the Wasserstein gradient flow. However, each step of the JKO scheme is costly as it requires solving a minimization problem involving the Wasserstein distance.

One way to compute the diffusion is to use a fixed discretization of the domain and apply standard numerical integration methods [18, 49, 15, 17, 40] to get $\rho_t$. For example, [50] proposes a method to approximate the diffusion based on JKO stepping and entropy-regularized optimal transport. However, these methods are limited to small dimensions since the discretization of space grows exponentially.

An alternative to domain discretization is particle simulation. It involves drawing random samples (particles) from the initial distribution and simulating their evolution via standard methods such as Euler-Maruyama scheme [36, §9.2]. After convergence, the particles are approximately distributed according to the stationary distribution, but no density estimate is readily available.

Another way to avoid discretization is to parameterize the density of $\rho_t$. Most methods approximate only the first and second moments $\rho_t$, e.g., via Gaussian approximation. Kalman filtering approaches can then compute the dynamics [34, 39, 33, 61]. More advanced Gaussian mixture approximations [65, 1] or more general parametric families have also been studied [64, 69]. In [48], variational methods are used to minimize the divergence between the predictive and the true density.

Recently, [24] introduced a parametric method to compute JKO steps via entropy-regularized optimal transport. The authors regularize the Wasserstein distance in the JKO step to ensure strict convexity and solve the unconstrained dual problem via stochastic program on a finite linear subset of basis functions. The method yields *unnormalized* probability density without direct sample access.

Recent works propose scalable continuous optimal transport solvers, parametrizing the solutions by reproducing kernels [10], fully-connected neural networks [62], or Input Convex Neural Networks (ICNNs) [37, 44, 38]. In particular, ICNNs gained attention for Wasserstein-2 transport since their gradients $\nabla \psi_\theta : \mathbb{R}^D \to \mathbb{R}^D$ can represent OT maps for the quadratic cost. These continuous solvers scale better to high dimension without discretizing the input measures, but they are too computationally expensive to be applied directly to JKO steps.

**Contributions.** We propose a scalable parametric method to approximate Wasserstein gradient flows via JKO stepping using input-convex neural networks (ICNNs) [6]. Specifically, we leverage Brenier's theorem to bypass the costly computation of the Wasserstein distance, and parametrize the optimal transport map as the gradient of an ICNN. Given sample access to the initial measure $\rho_0$, we use stochastic gradient descent (SGD) to sequentially learn time-discretized JKO dynamics of $\rho_t$. The trained model can sample from a continuous approximation of $\rho_t$ and compute its density $\frac{d\rho_t}{dx}(x)$. We compute gradient flows for the Fokker-Planck free energy functional $\mathcal{F}_{\text{FP}}$ given by (5), but our method generalizes to other cases. We demonstrate performance by computing diffusion following the Fokker-Planck equation and applying it to unnormalized density sampling as well as nonlinear filtering.

**Notation.** $\mathcal{P}_2(\mathbb{R}^D)$ denotes the set of Borel probability measures on $\mathbb{R}^D$ with finite second moment. $\mathcal{P}_{2,ac}(\mathbb{R}^D)$ denotes its subset of probability measures absolutely continuous with respect to Lebesgue measure. For $\rho \in \mathcal{P}_{2,ac}(\mathbb{R}^D)$, we denote by $\frac{d\rho}{dx}(x)$ its density with respect to the Lebesgue measure. $\Pi(\mu, \nu)$ denotes the set of probability measures on $\mathbb{R}^D \times \mathbb{R}^D$ with marginals $\mu$ and $\nu$. For measurable $T : \mathbb{R}^D \to \mathbb{R}^D$, we denote by $T\sharp$ the associated push-forward operator between measures.

## 2 Background on Wasserstein Gradient Flows

We consider gradient flows in Wasserstein space $(\mathcal{P}_2(\mathbb{R}^D), \mathcal{W}_2)$, the space of probability measures with finite second moment on $\mathbb{R}^D$ endowed with the Wasserstein-2 metric $\mathcal{W}_2$.

**Wasserstein-2 distance.** The (squared) Wasserstein-2 metric $\mathcal{W}_2$ between $\mu, \nu \in \mathcal{P}_2(\mathbb{R}^D)$ is

$$\mathcal{W}_2^2(\mu, \nu) \stackrel{\text{def}}{=} \min_{\pi \in \Pi(\mu, \nu)} \int_{\mathbb{R}^D \times \mathbb{R}^D} \|x - y\|_2^2 \, d\pi(x, y), \tag{1}$$

where the minimum is over measures $\pi$ on $\mathbb{R}^D \times \mathbb{R}^D$ with marginals $\mu$ and $\nu$ respectively [68].

For $\mu \in \mathcal{P}_{2,ac}(\mathbb{R}^D)$, there exists a $\mu$-unique map $\nabla\psi^* : \mathbb{R}^D \to \mathbb{R}^D$ that is the gradient of a convex function $\psi^* : \mathbb{R}^D \to \mathbb{R} \sqcup \{\infty\}$ satisfying $\nabla\psi^* \sharp \mu = \nu$ [46]. From Brenier's theorem [13], it follows that $\pi^* = [\mathrm{id}_{\mathbb{R}^D}, \nabla\psi^*]\sharp\mu$ is the unique minimizer of (1), i.e.,

$$\mathcal{W}_2^2(\mu, \nu) = \int_{\mathbb{R}^D} \|x - \nabla\psi^*(x)\|_2^2 \, d\mu(x).$$

**Wasserstein Gradient Flows.** In the Euclidean case, gradient flows along a function $f : \mathbb{R} \to \mathbb{R}$ follow the steepest descent direction and are defined through the ODE $\frac{dx_t}{dt} = -\boldsymbol{\nabla} f(x_t)$. Discretization of this flow leads to the gradient descent minimization algorithm. When functionals are defined over the space of measures equipped with the Wasserstein-2 metric, the equivalent flow is called the Wasserstein gradient flow. The idea is similar: the flow follows the steepest descent direction, but this time the notion of gradient is more complex. We refer the reader to [4] for exposition of gradient flows in metric spaces, or [59, Chapter 8] for an accessible introduction.

A curve of measures $\{\rho_t\}_{t \in \mathbb{R}_+}$ following the Wasserstein gradient flow of a functional $\mathcal{F}$ solves the continuity equation

$$\frac{\partial \rho_t}{\partial t} = \mathrm{div}(\rho_t \nabla_x \mathcal{F}'(\rho_t)), \qquad \text{s.t. } \rho_0 = \rho^0, \tag{2}$$

where $\mathcal{F}'(\cdot)$ is the first variation of $\mathcal{F}$ [4, Theorem 8.3.1]. The term on the right can be understood as the gradient of $\mathcal{F}$ in Wasserstein space, a vector field perturbatively rearranging the mass in $\rho_t$ to yield the steepest possible local change of $\mathcal{F}$.

Wasserstein gradient flows are used in various applied tasks. For example, gradient flows are applied in training [8, 43, 25] or refinement [7] of implicit generative models. In reinforcement learning, gradient flows facilitate policy optimization [55, 72]. Other tasks include crowd motion modelling [45, 58, 50], dataset optimization [2], and in-between animation [26].

Many applications come from the connection between Wasserstein gradient flows and SDEs. Consider an $\mathbb{R}^D$-valued stochastic process $\{X_t\}_{t \in \mathbb{R}_+}$ governed by the following Itô SDE:

$$dX_t = -\nabla\Phi(X_t)dt + \sqrt{2\beta^{-1}}dW_t, \qquad \text{s.t. } X_0 \sim \rho^0 \tag{3}$$

where $\Phi : \mathbb{R}^D \to \mathbb{R}$ is the potential function, $W_t$ is the standard Wiener process, and $\beta > 0$ is the magnitude. The solution of (3) is called an *advection-diffusion* process. The marginal measure $\rho_t$ of $X_t$ at each time satisfies the *Fokker-Planck equation* with fixed diffusion coefficient:

$$\frac{\partial \rho_t}{\partial t} = \mathrm{div}(\nabla\Phi(x)\rho_t) + \beta^{-1}\Delta\rho_t, \qquad \text{s.t. } \rho_0 = \rho^0. \tag{4}$$

Equation (4) is the Wasserstein gradient flow (2) for $\mathcal{F}$ given by the Fokker-Planck free energy functional [32]

$$\mathcal{F}_{\mathrm{FP}}(\rho) = \mathcal{U}(\rho) - \beta^{-1}\mathcal{E}(\rho), \tag{5}$$

where $\mathcal{U}(\rho) = \int_{\mathbb{R}^D} \Phi(x)d\rho(x)$ is the *potential energy* and $\mathcal{E}(\rho) = -\int_{\mathbb{R}^D} \log\frac{d\rho}{dx}(x)d\rho(x)$ is the *entropy*. As the result, to solve the SDE (3), one may compute the Wasserstein gradient flow of the Fokker-Planck equation with the free-energy functional $\mathcal{F}_{\mathrm{FP}}$ given by (5).

**JKO Scheme.** Computing Wasserstein gradient flows is challenging. The closed form solution is typically unknown, necessitating numerical approximation techniques. Jordan, Kinderlehrer, and Otto proposed a method—later abbreviated as *JKO integration*—to approximate the dynamics of $\rho_t$ in (2) [32]. It consists of a time-discretization update of the continuous flow given by:

$$\rho^{(k)} \leftarrow \underset{\rho \in \mathcal{P}_2(\mathbb{R}^n)}{\arg\min} \left[ \mathcal{F}(\rho) + \frac{1}{2h}\mathcal{W}_2^2(\rho^{(k-1)}, \rho) \right] \tag{6}$$

where $\rho^{(0)} = \rho^0$ is the initial condition and $h > 0$ is the time-discretization step size. The discrete time gradient flow converges to the continuous one as $h \to 0$, i.e., $\rho^{(k)} \approx \rho_{kh}$. The method was further developed in [4, 60], but performing JKO iterations remains challenging thanks to the minimization with respect to $\mathcal{W}_2$.

A common approach to perform JKO steps is to discretize the spatial domain. For support size $\lesssim 10^6$, (6) can be solved by standard optimal transport algorithms [51]. In dimensions $D \geq 3$, discrete supports can hardly approximate continuous distributions and hence the dynamics of gradient flows. To tackle this issue, [24] propose a stochastic parametric method to approximate the density of $\rho_t$. Their method uses entropy-regularized optimal transport (OT), which is biased.

# 3 Computing Wasserstein Gradient Flows with ICNNs

We now describe our approach to compute Wasserstein gradient flows via JKO stepping with ICNNs.

## 3.1 JKO Reformulation via Optimal Push-forwards Maps

Our key idea is to replace the optimization (6) over probability measures by an optimization over convex functions, an idea inspired by [11]. Thanks to Brenier's theorem, for any $\rho \in \mathcal{P}_{2,ac}$ there exists a unique $\rho^{(k-1)}$-measurable gradient $\nabla\psi : \mathbb{R}^D \to \mathbb{R}^D$ of a convex function $\psi$ satisfying $\rho = \nabla\psi \sharp \rho^{(k-1)}$. We set $\rho = \nabla\psi \sharp \rho^{(k-1)}$ and rewrite (6) as an optimization over convex $\psi$:

$$\psi^{(k)} \leftarrow \operatorname*{arg\,min}_{\text{Convex } \psi} \left[ \mathcal{F}(\nabla\psi \sharp \rho^{(k-1)}) + \frac{1}{2h} \mathcal{W}_2^2(\rho^{(k-1)}, \nabla\psi \sharp \rho^{(k-1)}) \right]. \tag{7}$$

To proceed to the next step of JKO scheme, we define $\rho^{(k)} \stackrel{\text{def}}{=} \nabla\psi^{(k)} \sharp \rho^{(k-1)}$.

Since $\rho$ is the pushforward of $\rho^{(k-1)}$ by the gradient of a convex function $\nabla\psi$, the $\mathcal{W}_2^2$ term in (7) can be evaluated explicitly, simplifying the Wasserstein-2 distance term in (7):

$$\psi^{(k)} \leftarrow \operatorname*{arg\,min}_{\text{Convex } \psi} \left[ \mathcal{F}(\nabla\psi \sharp \rho^{(k-1)}) + \frac{1}{2h} \int_{\mathbb{R}^D} \|x - \nabla\psi(x)\|_2^2 d\rho^{(k-1)}(x) \right]. \tag{8}$$

This formulation avoids the difficulty of computing Wasserstein-2 distances. An additional advantage is that we can *sample* from $\rho^{(k)}$. Since $\rho^{(k)} = [\nabla\psi^{(k)} \circ \cdots \circ \nabla\psi^{(1)}] \sharp \rho^0$, one may sample $x_0 \sim \rho^{(0)}$, and then $\nabla\psi^{(k)} \circ \cdots \circ \nabla\psi^{(1)}(x_0)$ gives a sample from $\rho^{(k)}$. Moreover, if functions $\psi^{(\cdot)}$ are strictly convex, then gradients $\nabla\psi^{(\cdot)}$ are invertible. In this case, the *density* $\frac{d\rho^{(k)}}{dx}$ of $\rho^{(k)} = \nabla\psi^{(k)} \circ \cdots \circ \nabla\psi^{(1)} \sharp \rho^0$ is computable by the change of variables formula (assuming $\psi^{(\cdot)}$ are twice differentiable)

$$\frac{d\rho^{(k)}}{dx}(x_k) = [\det \nabla^2 \psi^{(k)}(x_{k-1})]^{-1} \cdots [\det \nabla^2 \psi^{(1)}(x_0)]^{-1} \cdot \frac{d\rho^{(0)}}{dx}(x_0), \tag{9}$$

where $x_i = \nabla\psi^{(i)}(x_{i-1})$ for $i = 1, \ldots, k$ and $\frac{d\rho^{(0)}}{dx}$ is the density of $\rho^{(0)}$.

## 3.2 Stochastic Optimization for JKO via ICNNs

In general, the solution $\psi^{(k)}$ of (8) is intractable since it requires optimization over all convex functions. To tackle this issue, [11] discretizes the space of convex function. The approach also requires discretization of measures $\rho^{(k)}$ limiting this method to small dimensions.

We propose to parametrize the search space using input convex neural networks (ICNNs) [6] satisfying a universal approximation property among convex functions [20]. ICNNs are parametric models of the form $\psi_\theta : \mathbb{R}^D \to \mathbb{R}$ with $\psi_\theta$ convex w.r.t. the input. ICNNs are constructed from neural network layers, with restrictions on the weights and activation functions to preserve the input-convexity, see [6, §3.1] or [37, §B.2]. The parameters are optimized via deep learning optimization techniques such as SGD.

The JKO step then becomes finding the optimal parameters $\theta^*$ for $\psi_\theta$:

$$\theta^* \leftarrow \operatorname*{arg\,min}_{\theta} \left[ \mathcal{F}(\nabla\psi_\theta \sharp \rho^{(k-1)}) + \frac{1}{2h} \int_{\mathbb{R}^D} \|x - \nabla\psi_\theta(x)\|_2^2 d\rho^{(k-1)}(x) \right]. \tag{10}$$

If the functional $\mathcal{F}$ can be estimated stochastically using random batches from $\rho^{(k-1)}$, then SGD can be used to optimize $\theta$. $\mathcal{F}_{\text{FP}}$ given by (5) is an example of such a functional:

**Theorem 1** (Estimator of $\mathcal{F}_{\text{FP}}$). *Let $\rho \in \mathcal{P}_{2,ac}(\mathbb{R}^D)$ and $T : \mathbb{R}^D \to \mathbb{R}^D$ be a diffeomorphism. For a random batch $x_1, \ldots, x_N \sim \rho$, the expression $[\widehat{\mathcal{U}_T}(x_1, \ldots, x_N) - \beta^{-1}\widehat{\Delta\mathcal{E}_T}(x_1, \ldots, x_N)]$, where*

$$\widehat{\mathcal{U}_T}(x_1, \ldots, x_N) \stackrel{\text{def}}{=} \frac{1}{N} \sum_{n=1}^{N} \Phi\big(T(x_n)\big) \text{ and}$$

$$\widehat{\Delta\mathcal{E}_T}(x_1, \ldots, x_N) \stackrel{\text{def}}{=} \frac{1}{N} \sum_{n=1}^{N} \log|\det \nabla T(x_n)|,$$

*is an estimator of $\mathcal{F}_{FP}(T\sharp\rho)$ up to constant (w.r.t. $T$) shift given by $\beta^{-1}\mathcal{E}(\rho)$.*

*Proof.* $\widehat{\mathcal{U}_T}$ is a straightforward unbiased estimator for $\mathcal{U}(T\sharp\rho)$. Let $p$ and $p_T$ be the densities of $\rho$ and $T\sharp\rho$. Since $T$ is a diffeomorphism, we have $p_T(y) = p(x) \cdot |\det \nabla T(x)|^{-1}$ where $x = T^{-1}(y)$. Using the change of variables formula, we write

$$\mathcal{E}(T\sharp\rho) = -\int_{\mathbb{R}^D} p_T(y) \log p_T(y) dy$$

$$= -\int_{\mathbb{R}^D} p(x) \cdot |\det \nabla T(x)|^{-1} \log\left[p(x) \cdot |\det \nabla T(x)|^{-1}\right] \cdot |\det \nabla T(x)| dx$$

$$= -\int_{\mathbb{R}^D} p(x) \log p(x) dx + \int_{\mathbb{R}^D} p(x) \log |\det \nabla T(x)| dx$$

$$= \mathcal{E}(\rho) + \int_{\mathbb{R}^D} p(x) \log |\det \nabla T(x)| dx,$$

$$\implies \Delta\mathcal{E}_T(\rho) \overset{\text{def}}{=} \mathcal{E}(T\sharp\rho) - \mathcal{E}(\rho) = \int_{\mathbb{R}^D} \log |\det \nabla T(x)| d\rho(x)$$

which explains that $\widehat{\Delta\mathcal{E}_T}$ is an unbiased estimator of $\Delta\mathcal{E}_T(\rho)$. As the result, $\widehat{\mathcal{U}_T} - \beta^{-1}\widehat{\Delta\mathcal{E}_T}$ is an estimator for $\mathcal{F}_{\text{FP}}(T\sharp\rho) = \mathcal{U}(T\sharp\rho) - \beta^{-1}\mathcal{E}(T\sharp\rho)$ up to a shift of $\beta^{-1}\mathcal{E}(\rho)$. □

To apply Theorem 1 to our case, we take $T \leftarrow \nabla\psi_\theta$ and $\rho \leftarrow \rho^{(k-1)}$ to obtain a stochastic estimator for $\mathcal{F}_{\text{FP}}(\nabla\psi_\theta\sharp\rho^{(k-1)})$ in (10). Here, $\beta^{-1}\mathcal{E}(\rho^{(k-1)})$ is $\theta$-independent and constant since $\rho^{(k-1)}$ is fixed, so the offset of the estimator plays no role in the optimization w.r.t. $\theta$.

Algorithm 1 details our stochastic JKO method for $\mathcal{F}_{\text{FP}}$. The training is done solely based on random samples from the initial measure $\rho^0$: its density is not needed.

---

**Algorithm 1:** Fokker-Planck JKO via ICNNs

**Input** : Initial measure $\rho^0$ accessible by samples;
         JKO discretization step $h > 0$, number of JKO steps $K > 0$;
         target potential $\Phi(x)$, diffusion process temperature $\beta^{-1}$;
         batch size $N$;
**Output** : trained ICNN models $\{\psi^{(k)}\}_{k=1}^K$ representing JKO steps
**for** $k = 1, 2, \ldots, K$ **do**
     $\psi_\theta \leftarrow$ basic ICNN model;
     **for** $i = 1, 2, \ldots$ **do**
         Sample batch $Z \sim \rho^0$ of size N;
         $X \leftarrow \nabla\psi^{(k-1)} \circ \cdots \circ \nabla\psi^{(1)}(Z)$;
         $\widehat{\mathcal{W}_2^2} \leftarrow \frac{1}{N}\sum_{x \in X} \|\nabla\psi_\theta(x) - x\|_2^2$;
         $\widehat{\mathcal{U}} \leftarrow \frac{1}{N}\sum_{x \in X} \Phi(\nabla\psi_\theta(x))$;
         $\widehat{\Delta\mathcal{E}} \leftarrow \frac{1}{N}\sum_{x \in X} \log \det \nabla^2\psi_\theta(x)$;
         $\widehat{\mathcal{L}} \leftarrow \frac{1}{2h}\widehat{\mathcal{W}_2^2} + \widehat{\mathcal{U}} - \beta^{-1}\widehat{\Delta\mathcal{E}}$;
         Perform a gradient step over $\theta$ by using $\frac{\partial\widehat{\mathcal{L}}}{\partial\theta}$;
     $\psi^{(k)} \leftarrow \psi_\theta$

---

This algorithm assumes $\mathcal{F}$ is the Fokker-Planck diffusion energy functional. However, our method admits straightforward generalization to any $\mathcal{F}$ that can be stochastically estimated; studying such functionals is a promising avenue for future work.

### 3.3 Computing the Density of the Diffusion Process

Our algorithm provides a *computable density* for $\rho^{(k)}$. As discussed in §3.1, it is possible to sample from $\rho^{(k)}$ while simultaneously computing the density of the samples. However, this approach does not provide a direct way to evaluate $\frac{d\rho^{(k)}}{dx}(x_k)$ for arbitrary $x_k \in \mathbb{R}^D$. We resolve this issue below.

If a convex function is strongly convex, then its gradient is bijective on $\mathbb{R}^D$. By the change of variables formula for $x_k \in \mathbb{R}^D$, it holds $\frac{d\rho^{(k)}}{dx}(x_k) = \frac{d\rho^{(k-1)}}{dx}(x_{k-1}) \cdot [\det \nabla^2 \psi^{(k)}(x_{k-1})]^{-1}$ where $x_k = \nabla \psi^{(k)}(x_{k-1})$. To compute $x_{k-1}$, one needs to solve the convex optimization problem:

$$x_k = \nabla \psi^{(k)}(x_{k-1}) \qquad \Longleftrightarrow \qquad x_{k-1} = \arg\max_{x \in \mathbb{R}^D} \left[ \langle x, x_k \rangle - \psi^{(k)}(x) \right]. \tag{11}$$

If we know the density of $\rho^0$, to compute the density of $\rho^{(k)}$ at $x_k$ we solve $k$ convex problems

$$x_{k-1} = \arg\max_{x \in \mathbb{R}^D} \left[ \langle x, x_k \rangle - \psi^{(k)}(x) \right] \qquad \ldots \qquad x_0 = \arg\max_{x \in \mathbb{R}^D} \left[ \langle x, x_1 \rangle - \psi^{(1)}(x) \right]$$

to obtain $x_{k-1}, \ldots, x_0$ and then evaluate the density as

$$\frac{d\rho_k}{dx}(x_k) = \frac{d\rho^0}{dx}(x_0) \cdot \left[ \prod_{i=1}^{k} \det \nabla^2 \psi^{(i)}(x_{i-1}) \right]^{-1}.$$

Note the steps above provide a general method for tracing back the position of a particle along the flow, and density computation is simply a byproduct.

## 4 Experiments

In this section, we evaluate our method on toy and real-world applications. Our code is written in PyTorch and is publicly available at

> https://github.com/PetrMokrov/Large-Scale-Wasserstein-Gradient-Flows

The experiments are conducted on a GTX 1080Ti. In most cases, we performed several random restarts to obtain mean and variation of the considered metric. As the result, experiments require about 100-150 hours of computation. The details are given in Appendix A.

**Neural network architectures.** In all experiments, we use the DenseICNN [37, Appendix B.2] architecture for $\psi_\theta$ in Algorithm 1 with *SoftPlus* activations. The network $\psi_\theta$ is twice differentiable w.r.t. the input $x$ and has bijective gradient $\nabla \psi_\theta : \mathbb{R}^D \to \mathbb{R}^D$ with positive semi-definite Hessian $\nabla^2 \psi_\theta(x) \succeq 0$ at each $x$. We use automatic differentiation to compute $\nabla \psi_\theta$ and $\nabla^2 \psi_\theta$.

**Metric.** To qualitatively compare measures, we use the symmetric Kullback-Leibler divergence

$$\text{SymKL}(\rho_1, \rho_2) \overset{\text{def}}{=} \text{KL}(\rho_1 \| \rho_2) + \text{KL}(\rho_2 \| \rho_1), \tag{12}$$

where $\text{KL}(\rho_1 \| \rho_2) \overset{\text{def}}{=} \int_{\mathbb{R}^D} \log \frac{d\rho_1}{d\rho_2}(x) d\rho_1(x)$ is the Kullback-Leibler divergence. For particle-based methods, we obtain an approximation of the distribution by kernel density estimation.

### 4.1 Convergence to Stationary Solution

Starting from an arbitrary initial measure $\rho^0$, an advection-diffusion process (4) converges to the unique stationary solution $\rho^*$ [56] with density

$$\frac{d\rho^*}{dx}(x) = Z^{-1} \exp(-\beta \Phi(x)), \tag{13}$$

where $Z = \int_{\mathbb{R}^D} \exp(-\beta \Phi(x)) dx$ is the normalization constant. This property makes it possible to compute the symmetric KL between the distribution to which our method converges and the ground truth, provided $Z$ is known.

We use $\mathcal{N}(0, 16I_D)$ as the initial measure $\rho^0$ and a random Gaussian mixture as the stationary measure $\rho^*$. In our method, we perform $K = 40$ JKO steps with step size $h = 0.1$. We compare with a particle simulation method (with $10^3, 10^4, 10^5$ particles) based on the Euler-Maruyama $\lfloor$EM$\rfloor$ approximation [36, §9.2]. We repeat the experiment 5 times and report the averaged results in Figure 1.

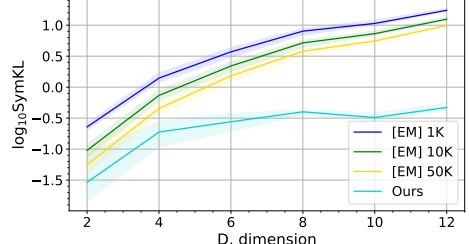

In Figure 2, we present qualitative results of our method converging to the ground truth in $D = 13, 32$.

Figure 1: SymKL between the computed and the stationary measure in $D = 2, 4, \ldots 12$

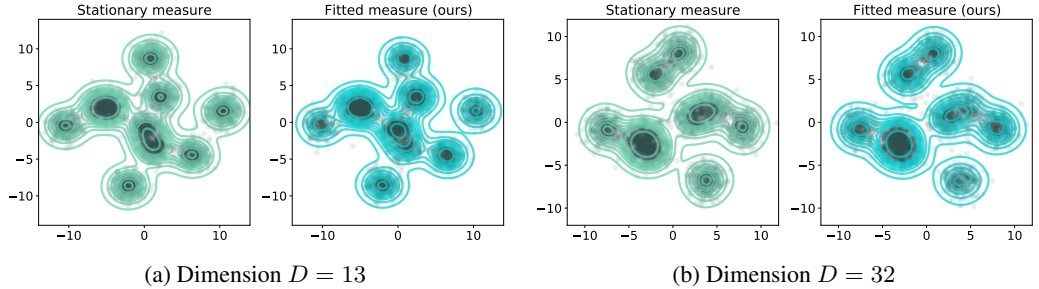

(a) Dimension $D = 13$        (b) Dimension $D = 32$

Figure 2: Projections to 2 first PCA components of the true stationary measure and the measure approximated by our method in dimensions $D = 13$ (on the left) and $D = 32$ (on the right).

## 4.2 Modeling Ornstein-Uhlenbeck Processes

Ornstein-Uhlenbeck processes are advection-diffusion processes (4) with $\Phi(x) = \frac{1}{2}(x-b)^T A(x-b)$ for symmetric positive definite $A \in \mathbb{R}^{D \times D}$ and $b \in \mathbb{R}^D$. They are among the few examples where we know $\rho_t$ for any $t \in \mathbb{R}^+$ in closed form, when the initial measure $\rho^0$ is Gaussian [67]. This allows to quantitatively evaluate the computed dynamics of the process, not just the stationary measure.

We choose $A, b$ at random and set $\rho^0$ to be the standard Gaussian measure $\mathcal{N}(0, I_D)$. We approximate the dynamics of the process by our method with JKO step $h = 0.05$ and compute SymKL between the true $\rho_t$ and the approximate one at time $t = 0.5$ and $t = 0.9$. We repeat the experiment 15 times in dimensions $D = 1, 2 \ldots, 12$ and report the performance at in Figure 3. The baselines are $\lfloor \text{EM} \rfloor$ with $10^3, 10^4, 5 \times 10^4$ particles, EM particle simulation endowed with the Proximal Recursion operator $\lfloor \text{EM PR} \rfloor$ with $10^4$ particles [16], and the parametric dual inference method [24] for JKO steps $\lfloor \text{Dual JKO} \rfloor$. The detailed comparison for times $t = 0.1, 0.2, \ldots 1$ is given in Appendix C.

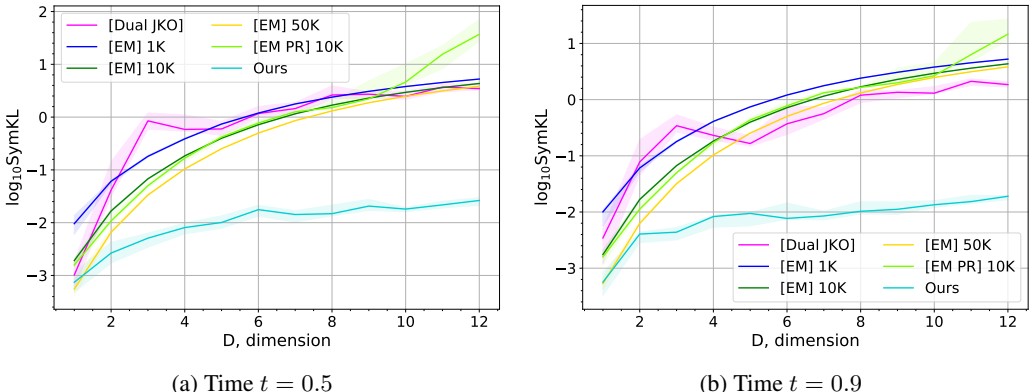

(a) Time $t = 0.5$        (b) Time $t = 0.9$

Figure 3: SymKL values between the computed measure and the true measure $\rho_t$ at $t = 0.5$ (on the left) and $t = 0.9$ (on the right) in dimensions $D = 1, 2, \ldots, 12$. Best viewed in color.

## 4.3 Unnormalized Posterior Sampling in Bayesian Logistic Regression

An important task in Bayesian machine learning to which our algorithm can be applied is sampling from an unnormalized posterior distribution. Given the model parameters $x \in \mathbb{R}^D$ with the prior distribution $p_0(x)$ as well as the conditional density $p(\mathcal{S}|x) = \prod_{m=1}^{M} p(s_m|x)$ of the data $\mathcal{S} = \{s_1, \ldots, s_M\}$, the posterior distribution is given by

$$p(x|\mathcal{S}) = \frac{p(\mathcal{S}|x)p_0(x)}{p(\mathcal{S})} \propto p(\mathcal{S}|x)p_0(x) = p_0(x) \cdot \prod_{m=1}^{M} p(s_m|x).$$

Computing the normalization constant $p(\mathcal{S})$ is in general intractable, underscoring the need for estimation methods that sample from $p(\mathcal{S}|x)$ given the density only up to a normalizing constant.

In our context, sampling from $p(x|\mathcal{S})$ can be solved similarly to the task in §4.1. From (13), it follows that the advection-diffusion process with temperature $\beta > 0$ and $\Phi(x) = -\frac{1}{\beta} \log \left[ p_0(x) \cdot p(\mathcal{S}|x) \right]$ has $\frac{d\rho^*}{dx}(x) = p(x|\mathcal{S})$ as the stationary distribution. Thus, we can use our method to approximate the diffusion process and obtain a sampler for $p(x|\mathcal{S})$ as a result.

The potential energy $\mathcal{U}(\rho) = \int_{\mathbb{R}^D} \Phi(x) d\rho(x)$ can be estimated efficiently by using a trick similar to the ones in stochastic gradient Langevin dynamics [70], which consists in resampling samples in $\mathcal{S}$ uniformly. For evaluation, we con-

| Dataset | Accuracy | | Log-Likelihood | |
|---|---|---|---|---|
| | Ours | ⌈SVGD⌋ | Ours | ⌈SVGD⌋ |
| covtype | 0.75 | 0.75 | -0.515 | -0.515 |
| german | 0.67 | 0.65 | -0.6 | -0.6 |
| diabetis | 0.775 | 0.78 | -0.45 | -0.46 |
| twonorm | 0.98 | 0.98 | -0.059 | -0.062 |
| ringnorm | 0.74 | 0.74 | -0.5 | -0.5 |
| banana | 0.55 | 0.54 | -0.69 | -0.69 |
| splice | 0.845 | 0.85 | -0.36 | -0.355 |
| waveform | 0.78 | 0.765 | -0.485 | -0.465 |
| image | 0.82 | 0.815 | -0.43 | -0.44 |

Table 1: Comparison of our method with ⌈SVGD⌋ [42] for Bayesian logistic regression.

sider the Bayesian linear regression setup of [42]. We use the 8 datasets from [47]. The number of features ranges from 2 to 60 and the dataset size from 700 to 7400 data points. We also use the Covertype dataset[2] with 500K data points and 54 features. The prior on regression weights $w$ is given by $p_0(w|\alpha) = \mathcal{N}(w|0, \alpha^{-1})$ with $p_0(\alpha) = \mathrm{Gamma}(\alpha|1, 0.01)$, so the prior on parameters $x = [w, \alpha]$ of the model is given by $p_0(x) = p_0(w, \alpha) = p_0(w|\alpha) \cdot p_0(\alpha)$. We randomly split each dataset into train $\mathcal{S}_{\mathrm{train}}$ and test $\mathcal{S}_{\mathrm{test}}$ ones with ratio 4:1 and apply the inference on the posterior $p(x|\mathcal{S}_{\mathrm{train}})$. In Table 1, we report accuracy and log-likelihood of the predictive distribution on $\mathcal{S}_{\mathrm{test}}$. As the baseline, we use particle-based Stein Variational Gradient Descent [42]. We use the author's implementation with the default hyper-parameters.

## 4.4 Nonlinear Filtering

We demonstrate the application of our method to filtering a nonlinear diffusion. In this task, we consider a diffusion process $X_t$ governed by the Fokker-Planck equation (4). At times $t_1 < t_2 < \cdots < t_K$ we obtain noisy observations of the process $Y_k = X_{t_k} + v_k$ with $v_k \sim \mathcal{N}(0, \sigma)$. The goal is to compute the predictive distribution $p_{t,X}(x|Y_{1:K})$ for $t \geq t_K$ given observations $Y_{1:K}$.

For each $k$ and $t \geq t_k$ predictive distribution $p_{t,X}(x|Y_{1:k})$ follows the diffusion process on time interval $[t_k, t]$ with initial distribution $p_{t_k,X}(x|Y_{1:k})$. If $t_k = t$ then

$$p_{t_k,X}(x|Y_{1:k}) \propto p(Y_k|X_{t_k} = x) \cdot p_{t_k,X}(x|Y_{1:k-1}). \tag{14}$$

For $k = 1, \ldots, K$, we sequentially obtain the predictive distribution $p_{t_k,X}(x|Y_{1:k})$ by using the previous predictive distribution $p_{t_{k-1},X}(x|Y_{1:k-1})$. First, given access to $p_{t_{k-1},X}(x|Y_{1:k-1})$, we approximate the diffusion on interval $[t_{k-1}, t_k]$ with initial distribution $p_{t_{k-1},X}(x|Y_{1:k-1})$ by our Algorithm 1 to get access to $p_{t_k,X}(x|Y_{1:k-1})$. Next, we use (14) to get unnormalized density and Metropolis-Hastings algorithm [57] to sample from $p_{t_k,X}(x|Y_{1:k})$. We give details in Appendix B.

For evaluation, we consider the experimental setup of [24, §6.3]. We assume that the 1-dimensional diffusion process $X_t$ has potential function $\Phi(x) = \frac{1}{\pi} \sin(2\pi x) + \frac{1}{4} x^2$ which makes the process highly nonlinear. We simulate nonlinear filtering on the time interval $t_{\mathrm{start}} = 0$ sec., $t_{\mathrm{fin}} = 5$ sec. and take the noise observations each 0.5 sec. The noise variance is $\sigma^2 = 1$ and $p(X_0) = \mathcal{N}(X_0|0, 1)$.

We predict the conditional density $p_{t_{\mathrm{final}},X}(x|Y_{1:9})$ and compare the prediction with ground truth obtained with numerical integration method by Chang and Cooper [19], who use a fine discrete grid. As the baselines, we use ⌊Dual JKO⌋ [24] as well as the Bayesian Bootstrap filter ⌊BBF⌋ [27], which combines particle simulation with bootstrap resampling at observation times.

We repeat the experiment 15 times. In Figure 4a, we report the SymKL between predicted density and true $p(X_{t_{\mathrm{fin}}}|Y_{1:9})$. We visually compare the fitted and true conditional distributions in Figure 4b.

## 5 Discussion

**Complexity of training and sampling.** Let $T$ be the number of operations required to evaluate ICNN $\psi_\theta(x)$, and assume that the evaluation of $\Phi(x)$ in the potential energy $\mathcal{U}$ takes $O(1)$ time.

---

[2]`https://www.csie.ntu.edu.tw/~cjlin/libsvmtools/datasets/binary.html`

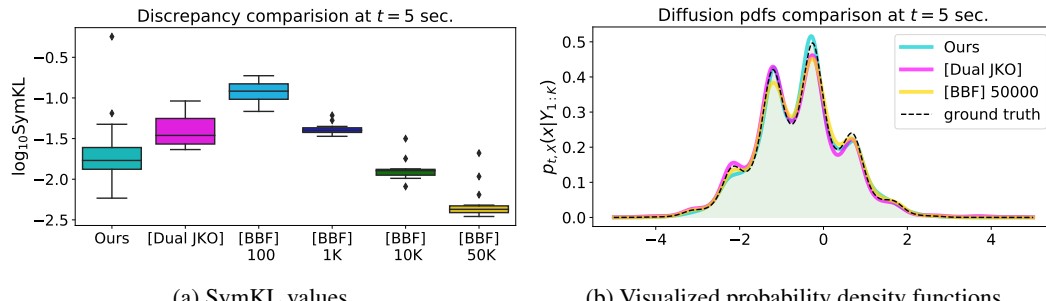

(a) SymKL values.

(b) Visualized probability density functions.

Figure 4: Comparison of the predicted conditional density and true $p(X_{t_{\text{fin}}}|Y_{1:9})$.

Recall that computing the gradient is a small constant factor harder than computing the function itself [41]. Thus, evaluation of $\nabla\psi_\theta(x) : \mathbb{R}^D \to \mathbb{R}^D$ requires $O(T)$ operations and evaluating the Hessian $\nabla^2\psi_\theta(x) : \mathbb{R}^D \to \mathbb{R}^{D\times D}$ takes $O(DT)$ time. To compute $\log\det\nabla^2\psi_\theta(x)$, we need $O(D^3)$ extra operations. Sampling from $\rho^{(k-1)} = \nabla\psi^{(k-1)} \circ \cdots \circ \nabla\psi^{(1)}\sharp\rho_0$ involves pushing $x_0 \sim \rho^0$ forward by a sequence of ICNNs $\psi^{(\cdot)}$ of length $k - 1$, requiring $O\big((k - 1)T\big)$ operations. The forward pass to evaluate the JKO step objective $\widehat{\mathcal{L}}$ in Algorithm 1 requires

| Operation | Time Complexity |
|---|---|
| Eval. $\psi_\theta, \nabla\psi_\theta, \nabla^2\psi_\theta$ | $T, O(T), O(DT)$ |
| Eval. $\log\det\nabla^2\psi_\theta$ | $O(DT+D^3)$ |
| Sample $x \sim \rho^{(k)}$ | $O((k-1)T)$ |
| Eval. $\widehat{\mathcal{L}}$ on $x \sim \rho^{(k)}$ | $O(DT + D^3)$ |
| Eval. $\frac{\partial\widehat{\mathcal{L}}}{\partial\theta}$ on $x \sim \rho^{(k)}$ | $O(DT+D^3)$ |
| Sample $x \sim \rho^{(k)}$ and Eval. $\frac{d\rho^{(k)}}{dx}(x)$ | $O\big((k-1)(TD+D^3)\big)$ |

Table 2: Complexity of operations in our method for computing JKO steps via ICNNs.

$O(DT + D^3)$ operations, as does the backward pass to compute the gradient $\frac{\partial\widehat{\mathcal{L}}}{\partial\theta}$ w.r.t. $\theta$.

The *memory complexity* is more difficult to characterize, since it depends on the autodiff implementation. It does not exceed the time complexity and is linear in the number of JKO steps $k$.

**Wall-clock times.** All particle-based methods considered in §4 and $\lfloor$Dual JKO$\rceil$ require from several seconds to several minutes CPU computation time. Our method requires from several minutes to few hours on GPU, the time is explained by the necessity to train a new network at each step.

**Advantages.** Due to using continuous approximation, our method scales well to high dimensions, as we show in §4.1 and §4.2. After training, we can produce infinitely many samples $x_k \sim \rho^{(k)}$, together with their trajectories $x_{k-1}, x_{k-2}, \ldots, x_0$ along the gradient flow. Moreover, the densities of samples in the flow $\frac{d\rho^{(k)}}{dx}(x_k), \frac{d\rho^{(k-1)}}{dx}(x_{k-1}), \ldots, \frac{d\rho^{(0)}}{dx}(x_0)$ can be evaluated immediately.

In contrast, particle-based and domain discretization methods do not scale well with the dimension (Figure 3) and provide no density. Interestingly, despite its parametric approximation, $\lfloor$Dual JKO$\rceil$ performs comparably to particle simulation and worse than ours (see additionally [24, Figure 3]).

**Limitations.** To train $k$ JKO steps, our method requires time proportional to $k^2$ due to the increased complexity of sampling $x \sim \rho^{(k)}$. This may be disadvantageous for training long diffusions. In addition, for very high dimensions $D$, exact evaluation of $\log\det\nabla^2\psi_\theta(x)$ is time-consuming.

**Future work.** To reduce the computational complexity of sampling from $\rho^{(k)}$, at step $k$ one may regress an invertible network $H : \mathbb{R}^D \to \mathbb{R}^D$ [9, 31] to satisfy $H(x_0) \approx \nabla\psi^{(k)} \circ \cdots \circ \nabla\psi^{(1)}(x_0)$ and use $H\sharp\rho_0 \to \rho^{(k)}$ to simplify sampling. An alternative is to use variational inference [12, 54, 71] to approximate $\rho^{(k)}$. To mitigate the computational complexity of computing $\log\det\nabla^2\psi_\theta(x)$, fast approximation can be used [66, 28]. More broadly, developing ICNNs with easily-computable exact Hessians is a critical avenue for further research as ICNNs continue to gain attention in machine learning [44, 37, 38, 30, 23, 5].

**Potential impact.** Diffusion processes appear in numerous scientific and industrial applications, including machine learning, finances, physics, and population dynamics. Our method will improve models in these areas, providing better scalability. Performance, however, might depend on the expressiveness of the ICNNs, pointing to theoretical convergence analysis as a key topic for future study to reinforce confidence in our model.

In summary, we develop an efficient method to model diffusion processes arising in many practical tasks. We apply our method to common Bayesian tasks such as unnormalized posterior sampling (§4.3) and nonlinear filtering (§4.4). Below we mention several other potential applications:

- **Population dynamics.** In this task, one needs to recover the potential energy $\Phi(x)$ included in the Fokker-Planck free energy functional $\mathcal{F}_{\text{FP}}$ based on samples from the diffusion obtained at timesteps $t_1, \ldots, t_n$, see [29]. This setting can be found in computational biology, see §6.3 of [29]. A recent paper [14] utilizes ICNN-powered JKO to model population dynamics.
- **Reinforcement learning**. Wasserstein gradient flows provide a theoretically-grounded way to optimize an agent policy in reinforcement learning, see [55, 72]. The idea of the method is to maximize the expected total reward (see (10) in [72]) using the gradient flow associated with the Fokker-Planck functional (see (12) in [72]). The authors of the original paper proposed discrete particle approximation method to solve the underlying JKO scheme. Substituting their approach with our ICNN-based JKO can potentially improve the results.
- **Refining Generative Adversarial Networks**. In the GAN setting, given trained generator $G$ and discriminator $D$, one can improve the samples from $G$ by $D$ via considering a gradient flow w.r.t. entropy-regularized $f$-divergence between real and generated data distribution (see [7], in particular, formula (4) for reference). Using KL-divergence makes the gradient flow consistent with our method: the functional $\mathcal{F}$ defining the flow has only entropic and potential energy terms. The usage of our method instead of particle simulation may improve the generator model.
- **Molecular Discovery.** In [3], in parallel to our work the JKO-ICNN scheme is proposed. The authors consider the molecular discovery as an application. The task is to increase the *drug-likeness* of a given distribution $\rho$ of molecules while staying close to the original distribution $\rho_0$. The task reduces to optimizing the functional $\mathcal{F}(\rho) = \mathbb{E}_{x \sim \rho} \Phi(x) + \mathcal{D}(\rho, \rho_0)$ for a certain potential $\Phi$ ($V$ - in the notation of [3]) and a discrepancy $\mathcal{D}$. The authors applied the JKO-ICNN method to minimize $\mathcal{F}$ on MOSES [53] molecular dataset and obtained promising results.

ACKNOWLEDGEMENTS. Skoltech acknowledges the support of the Ministry of Science and Higher Education grant No. 075-10-2021-068. The MIT Geometric Data Processing group acknowledges the generous support of Army Research Office grants W911NF2010168 and W911NF2110293, of Air Force Office of Scientific Research award FA9550-19-1-031, of National Science Foundation grants IIS-1838071 and CHS-1955697, from the CSAIL Systems that Learn program, from the MIT–IBM Watson AI Laboratory, from the Toyota–CSAIL Joint Research Center, from a gift from Adobe Systems, from an MIT.nano Immersion Lab/NCSOFT Gaming Program seed grant, and from the Skoltech–MIT Next Generation Program.

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
