# A Experimental Details

**General details.** We use DenseICNN architecture [37, Appendix B.2] for $\psi_\theta$ with 2 hidden layers and vary the width of the model depending on the task. We use Adam optimizer with learning rate decreasing with the number of JKO steps. We initialize the ICNN models either via pretraining to satisfy $\nabla\psi_\theta(x) \approx x$ or by using parameters $\theta$ obtained from the previous JKO step.

For $\lfloor$Dual JKO$\rfloor$, we used the implementation provided by the authors with default hyper-parameters. For $\lfloor$EM PR$\rfloor$ we implemented the Proximal Recursion operator following the pseudocode of [16] and used the default hyper-parameters but we increased the number of particles for fair comparison with the vanilla $\lfloor$EM$\rfloor$ algorithm. Note we limited the number of particles to $N = 10^4$ because of the high computational complexity of the method. For $\lfloor$SVGD$\rfloor$, we used the official implementation available at

$$\texttt{https://github.com/dilinwang820/Stein-Variational-Gradient-Descent}$$

In particle-based simulations $\lfloor$EM$\rfloor$, $\lfloor$BBF$\rfloor$ and $\lfloor$EM PR$\rfloor$ we used the particle propagation timestep $dt = 10^{-3}$.

We estimate the SymKL (12) using Monte Carlo (MC) on $10^4$ samples. In our method, MC estimate is straightforward since the method permits both sampling and computing the density. In particle-based methods, we use kernel density estimator to approximate the density utilizing `scipy` implementation of `gaussian_kde` with `bandwidth` chosen by Scott's rule. In $\lfloor$Dual JKO$\rfloor$, we employ importance sampling procedure and normalization constant estimation as detailed in [24].

We set $\beta$ to be equal to 1 throughout our experiments.

## A.1 Converging to Stationary Distribution

As the stationary measure $\rho^*$ we consider random Gaussian mixture $\frac{1}{N_p}\sum_{m=1}^{M}\mathcal{N}(\mu_m, I_D)$, where $\mu_1, \ldots, \mu_M \sim \text{Uniform}\left([-\frac{l}{2}, \frac{l}{2}]^D\right)$. We set the width $w$ of used ICNNs $\psi_\theta$ depending on dimension $D$. The parameters are summarized in Table 3.

Each JKO step uses 1000 gradient descent iterations of Algorithm 1. For dimensions $D = 2, 4, \ldots, 12$ the first 20 JKO transitions are optimized with $lr = 5 \cdot 10^{-3}$ and the remaining steps use $lr = 2 \cdot 10^{-3}$. For qualitative experiments in $D = 13, 32$ we perform 50 and 70 JKO steps with step size $h = 0.1$. The learning rate setup in these cases is similar to quantitative experiment setting but has additional stage with $lr = 5 \cdot 10^{-4}$ on the final JKO steps. The batch size is $N = 512$.

| $D$ | $M$ | $l$ | $w$ |
|---|---|---|---|
| 2 | 5 | 10 | 256 |
| 4 | 6 | 10 | 384 |
| 6 | 7 | 10 | 512 |
| 8 | 8 | 10 | 512 |
| 10 | 9 | 10 | 512 |
| 12 | 10 | 10 | 1024 |
| 13 | 10 | 10 | 512 |
| 32 | 10 | 6 | 1024 |

Table 3: Hyper-parameters in the convergence exp.

## A.2 Modeling Ornshtein-Uhlenbeck Processes

Matrices $A \in \mathbb{R}^{D \times D}$ are randomly generated using `sklearn.datasets.make_spd_matrix`. Vectors $b \in \mathbb{R}^D$ are sampled from standard Gaussian measure. All ICNNs $\psi_\theta$ have $w = 64$ and we train each of them for 500 iterations per JKO step with $lr = 5 \cdot 10^{-3}$ and batch size $N = 1024$.

## A.3 Unnormalized Posterior Sampling

To remove positiveness constraint on $\alpha$ we consider $[w, \log(\alpha)]$ as the regression model parameters instead of $[w, \alpha]$. To learn the posterior distribution $p(x|S_{\text{train}})$ we use JKO step size $h = 0.1$. Let $iter$ denote the number of gradient steps over $\theta$ per each JKO step. The used hyper-parameters for each dataset are summarized in Table 4.

To estimate the log-likelihood and accuracy of the predictive distribution on $S_{test}$ based

| Dataset | $w$ | $lr$ | $iter$ | batch | $K$ |
|---|---|---|---|---|---|
| covtype | 512 | $2 \cdot 10^{-5}$ | $10^4$ | 1024 | 6 |
| german | 512 | $2 \cdot 10^{-4}$ | 5000 | 512 | 5 |
| diabetis | 128 | $5 \cdot 10^{-5}$ | 6000 | 1024 | 16 |
| twonorm | 512 | $5 \cdot 10^{-5}$ | 5000 | 1024 | 7 |
| ringnorm | 512 | $5 \cdot 10^{-5}$ | 5000 | 1024 | 2 |
| banana | 128 | $2 \cdot 10^{-4}$ | 5000 | 1024 | 5 |
| splice | 512 | $2 \cdot 10^{-3}$ | 2000 | 512 | 5 |
| waveform | 512 | $5 \cdot 10^{-5}$ | 5000 | 512 | 2 |
| image | 512 | $5 \cdot 10^{-5}$ | 5000 | 512 | 5 |

Table 4: Hyper-parameters we use in Bayesian logistic regression experiment.

on $p(x|S_{\text{train}})$, we use straightforward MC estimate on $2^{12}$ random parameter samples.

## B  Nonlinear Filtering Details

For $k = 1, 2, \ldots$ we progressively obtain access to samples (and their un-normalized density) from predictive distribution $p_{t_k, X}(x|Y_{1:k})$ for step $k$ given $k$ observations $Y_1, \ldots, Y_k$.

First, at each step $k$, we access $p_{t_k, X}(x|Y_{1:k})$ through $p_{t_{k-1}, X}(x|Y_{1:k-1})$. To do this, we use our Algorithm 1 to model a diffusion on $[t_{k-1}, t_k]$ with initial distribution $p_{t_{k-1}, X}(x|Y_{1:k-1})$. We perform $n_k$ JKO steps of size $h_k = \frac{t_k - t_{k-1}}{n_k}$ and obtain ICNNs $\psi_1^{(k)}, \ldots, \psi_{n_k}^{(k)}$ (approximately) satisfying

$$\mu_{p_{t_k, X}(x|Y_{1:k-1})} = [\nabla \psi_{n_k}^{(k)} \circ \cdots \circ \nabla \psi_1^{(k)}] \sharp \mu_{p_{t_{k-1}, X}(x|Y_{1:k-1})} \tag{15}$$

Here $\mu_{p(\cdot)}$ is the measure with density $p(\cdot)$. We define $B_k \stackrel{def}{=} \nabla \psi_{n_k}^{(k)} \circ \cdots \circ \nabla \psi_1^{(k)}$.

Let $x_k \in \mathbb{R}^D$ and sequentially define $x_{i-1} = B_i^{-1}(x_i)$ for $i = k, \ldots, 1$. We derive

$$p_{t_k, X}(x_k|Y_{1:k}) \stackrel{(14)}{\propto}$$

$$p(Y_k|X_{t_k} = x_k) \cdot p_{t_k, X}(x_k|Y_{1:k-1}) \stackrel{(15)}{=}$$

$$p(Y_k|X_{t_k} = x_k) \cdot [\det \nabla B_k(x_{k-1})]^{-1} \cdot p_{t_{k-1}, X}(x_{k-1}|Y_{1:k-1}) \stackrel{(14)}{\propto}$$

$$\cdots$$

$$\prod_{i=1}^{k} p(Y_i|X_{t_i} = x_i) \cdot [\prod_{i=1}^{k} \det \nabla B_i(x_{i-1})]^{-1} \cdot p_{t_0, X}(x_0) \tag{16}$$

where we substitute (14) sequentially for $k, k-1, \ldots, 1$. As the result, from (16) we obtain the unnormalized density of predictive distribution $p_{t_k, X}(x_k|Y_{1:k})$. To sample from the predictive distribution (to train the next step $k+1$) we use Metropolis-Hastings algorithm [57]. For completeness, we recall the algorithm 2 below. The algorithm builds a chain $x^{(1)}, x^{(2)}, \ldots$ converging to the distribution given by unnormalized density $\pi(\cdot)$. As input, the algorithm also takes a family of proposal distributions $q_x(\cdot)$ for $x \in \mathbb{R}^D$. The value $\alpha(\cdot, \cdot)$ is called the acceptance probability.

---

**Algorithm 2:** Metropolis-Hastings algorithm

---

**Input** :Unnormalized density $\pi(\cdot)$; family of proposal distributions $q_x(\cdot)$ ($x \in \mathbb{R}^D$
**Output** :Sequence $x^{(1)}, x^{(2)}, x^{(3)}, \ldots$ of samples from $\pi$
Select $x^{(0)} \in \mathbb{R}^D$
**for** $j = 1, 2, \ldots$ **do**
  Sample $y \sim q_{x^{(j-1)}}$;
  Compute $\alpha(x^{(j-1)}, y) = \min \left( 1, \frac{\pi(y) q_y(x^{(j-1)})}{\pi(x^{(j-1)}) q_{x^{(j-1)}}(y)} \right)$
  With probability $\alpha(x^{(j-1)}, y)$ set $x^{(j)} \leftarrow y$; otherwise set $x^{(j)} \leftarrow x^{(j-1)}$

---

To sample from $p_{t_k, X}(x_k|Y_{1:k})$ we use Algorithm 2 with $\pi$ equal to unnormalized density (16). We note that computing $\pi(x_k)$ for $x_k \in \mathbb{R}^D$ is not easy since it requires computing pre-images $x_{k-1}, \ldots, x_0$ by inverting $B_k, B_{k-1}, \ldots, B_1$. As the consequence, this makes computation of acceptance probability $\alpha(\cdot, \cdot)$ hard. To resolve this issue, we choose special $x$-independent proposals

$$q = q_x \stackrel{\text{def}}{=} (B_k \circ B_{k-1} \circ \cdots \circ B_1) \sharp \mu_{p_0, X}. \tag{17}$$

In this case, all $\det$ terms in $\alpha(x, y)$ vanish simplifying the computation (we write $x = x_k, y = y_k$):

$$\frac{\pi(y) q_y(x)}{\pi(x) q_x(y)} = \frac{\pi(y) q(x)}{\pi(x) q(y)} =$$

$$\frac{p_{0,X}(y_0) \prod_{i=1}^{k} p_{t_i,Y}(Y_i|X_{t_i} = y_i) \prod_{i=1}^{k} \det \nabla B_i(x_{i-1}) \cdot p_{0,X}(x_0) \prod_{i=1}^{k} \det \nabla B_i(y_{i-1})}{p_{0,X}(x_0) \prod_{i=1}^{k} p_{t_i,Y}(Y_i|X_{t_i} = x_i) \prod_{i=1}^{k} \det \nabla B_i(y_{i-1}) \cdot p_{0,X}(y_0) \prod_{i=1}^{k} \det \nabla B_i(x_{i-1})} =$$

$$\frac{\prod_{i=1}^{k} p_{t_i,Y}(Y_i|X_{t_i} = y_i)}{\prod_{i=1}^{k} p_{t_i,Y}(Y_i|X_{t_i} = x_i)} \quad (18)$$

To compute (18) one needs to know preimages $x_{k-1}, \ldots, x_0$ and $y_{k-1}, \ldots, y_0$ of points $y = y_k$ and $x = x_k$ respectively. They can be straightforwardly computed when sampling from $q$ happens (17).

**Experimental details.** To obtain the noise observations $Y_k = X_{t_k} + v_k$ from the process, we simulate a particle $X_0$ randomly sampled from the initial measure $\mathcal{N}(0,1)$ by using Euler-Maruyama method to obtain the trajectory $X_t$. At observation times $t_1 = 0.5, \ldots, t_9 = 4.5$ we add random noise $v_k \sim \mathcal{N}(0,1)$ to obtain observations $Y_1, \ldots, Y_9$.

We utilize Chang and Cooper [19] numerical integration method to compute true $p(X_{t_{\text{fin}}}|Y_{1:9})$. We construct regular fine grid on the segment $[-5, 5]$ with 2000 points and numerically solve the SDE with timestep $dt = 10^{-3}$. At observation times $t_k$, $k \in 1, \ldots 9$ we multiply the obtained probability density function $p_{t_k,X}(x|Y_{1:k-1})$ by the density of the normal distribution $p(Y_k|X_{t_k} = x)$ estimated at the grid which results in unnormalized $p_{t_k,X}(x|Y_{1:k})$. After normalization on the grid, $p_{t_k,X}(x|Y_{1:k})$ can be used in the new diffusion round on time interval $[t_k, t_{k+1}]$. At final time $t_{\text{fin}}$ we estimate SymKL between the true distribution and ones obtained via other competitive methods by numerically integrating (12) on the grid.

We implement $\lfloor$BBF$\rfloor$ following the original article [27]. Particle propagation performed via Euler-Maruyama method with timestep $dt = 10^{-3}$. The final distribution $p(X_{t_{\text{fin}}}|Y_{1:9})$ is estimated using kernel density estimator as described in Appendix A.

For $\lfloor$Dual JKO$\rfloor$ we use the code provided by the authors with the default hyper-parameters.

In our method, we use JKO step size $h = 0.1$ and model it by ICNN with width $w = 256$. Each JKO step takes 700 optimization iterations with $lr = 5 \cdot 10^{-3}$ and batch size $N = 1024$. At observation times $t_k$, $k \in 1, 2, \ldots 9$ we use the Metropolis-Hastings algorithm 2 with acceptance probability $\alpha$ calculated by (18). Starting from the randomly sampled $x^{(1)}$ we skip the first 1000 values of the Markov Chain generated by the algorithm which allows the series to converge to the distribution of interest $p_{t_k,X}(x|Y_{1:k})$. We take each second element from the chain in order to decorrelate the samples. To simultaneously sample the batch of size $N$, we run $N$ chains in parallel. To compute SymKL, we normalize the resulting distribution $p(X_{t_{\text{fin}}}|Y_{1:9})$ on the Chang-Cooper support grid.

## C  Additional Experiments

In Figure 5, we compare the true distribution $\rho_t$ with the predicted distribution via the competitive methods when modelling Ornstein-Uhlenbeck processes (§4.2). The comparison is given for time $t = 0.1, 0.2, \ldots, 1.0$.

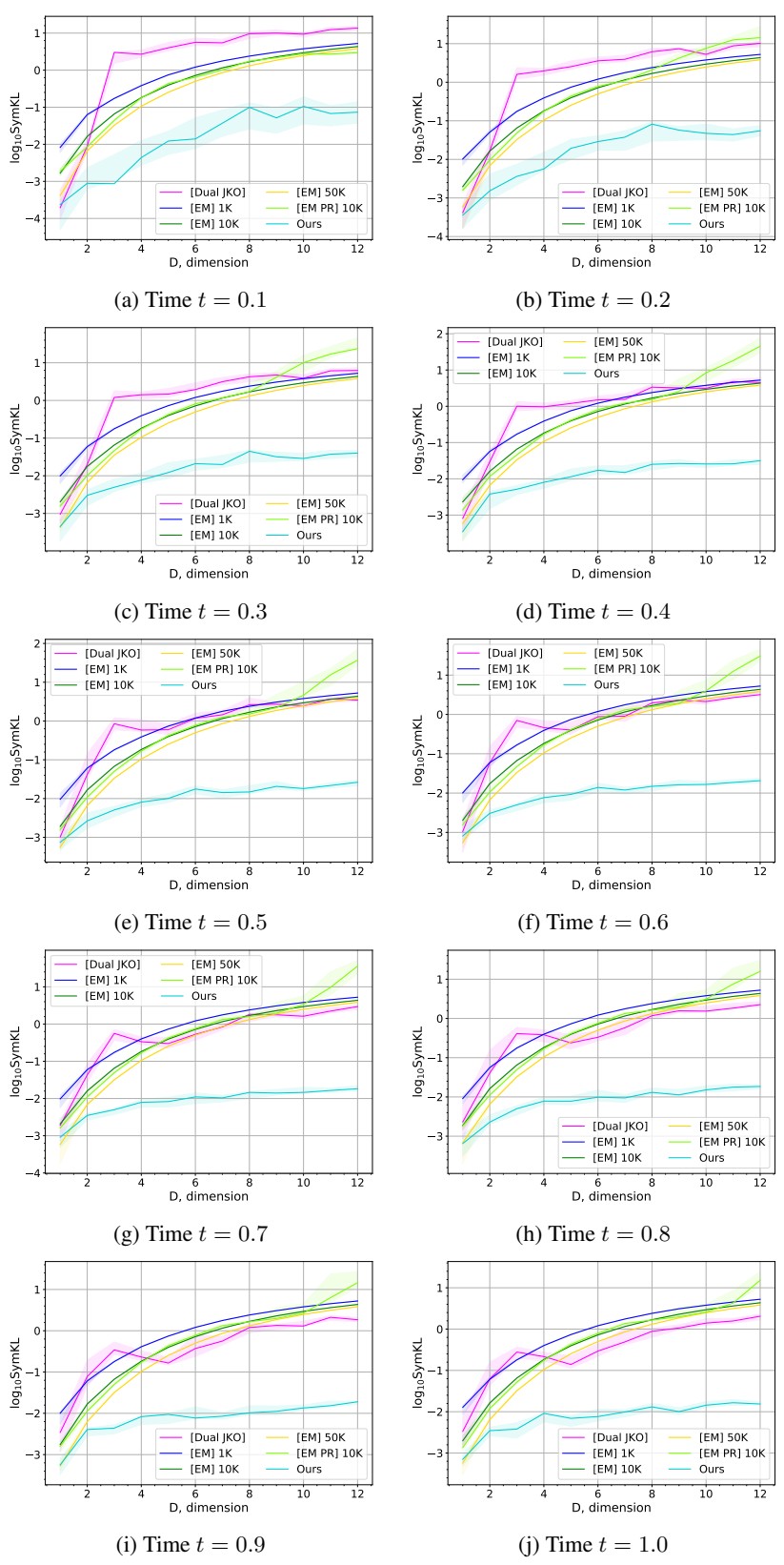

Figure 5: SymKL values between the computed measures and the true measure at $t = 0.1, 0.2, \ldots, 1$ in dimensions $D = 1, 2, \ldots 12$. Best viewed in color.