# OpenReview forum: "Large-Scale Wasserstein Gradient Flows"
_NeurIPS.cc/2021/Conference — NeurIPS 2021 Poster_

### Official Review · Reviewer_yWvM · 2021-07-15

**Rating:** 6
**Confidence:** 4

**Summary:**

This paper proposes an input-convex neural networks (ICNNs) based method to approximate the JKO scheme for discretizing Wasserstein gradient flows. Following a previous work [1], the key idea is to reformulate the distribution optimization problem in each JKO iteration as optimization over convex functions due to the well-known Brenner’s polar factorization theorem [2]. ICNNs are used to parameterize these convex functions from computational considerations. The proposed method is applied to simulate Wasserstein gradient flows for sampling from unnormalized density and nonlinear filtering.
[1] Discretization of functionals involving the Monge-Ampere operator.
[2] Polar factorization and monotone rearrangement of vector-valued functions.


**Ethical Concerns:**

No.

**Limitations And Societal Impact:**

No.

**Main Review:**

It is a novel idea to introduce ICNNs for addressing the computational challenges of the JKO scheme. ICNNs may be a promising tool to make the JKO scheme feasible for large-scale computations.
Quality: The work is technically sound.
Pros: The methodology part is well-organized, and the advance of the proposed method is clear.
The potential applications are interesting, such as sampling from unnormalized density, nonlinear filtering.
Cons: (1) Maybe the title is a little ambiguous. It would be better to include input-convex neural networks (ICNNs) in the title since the proposed method is entirely based on ICNNs.
(2) As the title indicates, the focus of this work is a large-scale setting. However, it is hard to conclude that the large-scale setting is well considered in the experimental part. For example, the considered data dimension is at most 32 in Section 4.1, 12 in Section 4.2, 60 in Section 4.3, 1 in Section 4.4.



**Time Spent Reviewing:**

5

---

> ### Author Response · Authors · 2021-08-07
> **Answers to Reviewer yWvM**
>
> Dear reviewer, thanks for your thoughtful review. Please find below the answers to your comments and questions.
>
> **(1) Paper title (large scale).**
>
> The majority of existing work for modeling gradient flows, e.g. [10,46], consider dimensions $\leq 3$. Therefore, w.r.t. those works, our method considers large scale diffusions at higher dimensions.
>
> **(2) Paper title (ICNNs).**
>
> We did not add "Input convex neural networks" to the title to keep it simple.
> Additionally, our method is generic for other convex neural networks. At the time of writing, the ICNN is still the go-to choice for large-scale convex paramaterization in machine learning.
>
> **Concluding remarks.**
> Please respond to our post to let us know if the clarifications above suitably address your concerns about our work.  We are happy to address any remaining points during the discussion phase; if the responses above are sufficient, we kindly ask that you consider raising your score.

---

> > ### Comment · Reviewer_yWvM · 2021-08-23
> > **Thanks for the authors reply.**
> >
> > Thanks for the authors' reply. I will keep my score unchanged.

---

### Official Review · Reviewer_tTtz · 2021-07-16

**Rating:** 6
**Confidence:** 5

**Summary:**

This paper is on the computation of Wasserstein gradient flow, a type of gradient flow based on the optimal transport theory. The proposed algorithm is built on the famous JKO scheme. To implement the JKO scheme, input-convex neural networks (ICNNs) are used to model the optimal transport map between the distributions at consecutive steps.

**Limitations And Societal Impact:**

Limitations and societal impact are properly discussed.

**Main Review:**

Originality:
Computing Wasserstein gradient flow is a classical problem in optimal transport. Most existing algorithms requires discretization of the underlying space and are thus not scalable. The present work starts from the JKO scheme, an implicit scheme, for Wasserstein gradient flow, and utilizes ICNNs to realize a JKO step. The objective free energy is also estimated efficiently using samples as in Theorem 1. The combination of these two techniques is new. A related work on particle-based Fokker-planck is [1].

Quality:
The main results on the Wasserstein gradient flow of the Fokker-planck free energy are technically sound.
1. Since this gradient flow is mainly for sampling purpose, the authors can include more comprehensive comparison with more advanced sampling algorithms other than overdamped Langevin (EM).
2. There is only one example (4.2) to illustrate the temporal behavior of the gradient flow and only two time points (t=0.5, 0.9) are evaluated. More examples are needed.
3. The claim ``our method admits straightforward generalization to any F that ….,’’ should either be removed or more properly explained. The proposed algorithm does use some special property of the Fokker-planck free energy that cannot be generalized.
4. The results of dual JKO in figure 4 seems to be in consistent with [2].
5. The algorithm seems to have larger variance than EM, especially for low dimensional problems (Figure 1, 3, 4).

Clarity:
The paper is over-all well-written.
1. The experiments can be improved by adding more details. For instance, how is Accuracy in Table 1 defined?
2. The notation T is used in Theorem 1 and Section 5 for different purposes.
3. It should be argmax in (11) and the following equations

Significance:
The computation of Wasserstein gradient flow is an important problem. The proposed algorithm provides a possible way to calculate Wasserstein gradient flow in high dimensional setting. One limitation of this algorithm is that it is only applicable to the Fokker-planck free energy.

[1] K.F. Caluya et al, Proximal recursion for solving the Fokker-Planck equation
[2] C. Frogner et al, Approximate inference with Wasserstein gradient flows

**Time Spent Reviewing:**

3

---

> ### Author Response · Authors · 2021-08-07
> **Answers to Reviewer tTtz**
>
> Dear reviewer,
>
> Thanks for your thoughtful review. We will improve the clarity according to your comments. We thank you for pointing to the particle simulation method by Caluya et al. (2018); we will add the citation. Please find below the answers to your comments and questions.
>
> **(1) Advanced sampling algorithms.**
>
> In our view, the issue of particle simulation in high dimensions is not in the procedure to simulate evolution, but in the fact that the evolving distribution is being discretized. In high dimensions, this strategy suffers from the curse of dimensionality. Moreover, kernel density estimation needed to recover SymKL values itself is known to perform poorly in high dimensions. Therefore, comparison with other particle simulation or domain discretization methods will not change the overall picture.
>
> **(2) Temporal behavior of the gradient flow.**
>
> Due to space limitations, in Section 4.2, we indeed only include results for two time steps $t=0.5$ and $t=0.9$. You can find the results for more time steps in the supplementary material in *results/ou\_vary\_dim/ou\_vary\_dim\_freq-ICNN\_jko#.json* files with SymKL for all our experiments of Section 4.2. Metric values are stored for $t=[0.1, 0.2, \dots,0.9, 1.0]$ and can be used for producing plots similar to Figure 3 for other time steps $t$. However, we decided not to include these plots because they anyway show the same trends as Figure 3: our method outperforms the competing methods. We will add these plots to the Appendix in the final version.
>
> **(3) Extensions to other diffusions.**
>
> To use functional $\mathcal{F}$ with our method, we need to be able to stochastically estimate $\mathcal{F}(T\sharp\rho)$. For Fokker-Plank free energy functional, we provided a way to do this (Theorem 1) by employing special property of entropy $\mathcal{E}$. Nevertheless, for potential energy $\mathcal{U}$ (line 95), we used a straightforward Monte-Carlo estimator (lines 141-142) which can be easily generalized to other functionals.
>
> For example, interaction energy functional $\mathcal{I}(\rho)=\int W(x-y)d\rho(x)d\rho(y)$ also appears in many practical tasks. Similar to potential energy $\mathcal{U}$, the value $\mathcal{I}(T\sharp\rho)$ admits estimates from batches $x_i\sim\rho$ and $y_j\sim\rho$, i.e. $\mathcal{I}(T\sharp\rho)\approx \frac{1}{N^{2}}\sum_{i=1}^{N}\sum_{j=1}^{N}W(T(x_{i})-T(y_{j}))$.
>
> The list of suitable functionals $\mathcal{F}(\rho)$ is much wider, see [10, eq. (1.9-1.11)] for more examples. We will add this reference to lines 153-155. Studying applications of our method to these functionals is a promising avenue for future work.
>
> **(4) Inconsistent results with [21] (Frogner et al. (2020)).**
>
> To reproduce the method of [21], we used the code provided by the authors (line 513 of Appendix). However, in the nonlinear filtering experiment (Section 4.4) we adopt the different SymKL evaluation scheme. The authors of [21] reported the average of SymKL for posterior distributions $p_{t_k, X}(x | Y_{1:k}) $, at all observation times, while we used only the SymKL for the posterior distribution at the last step of the diffusion.
> Predicting the last step is harder than intermediate steps due to the accumulation of the error. Besides, for particle simulation method, we consider $1K$, $10K$, $50K$ particles, while in [21, Figure 3b] sizes $0.1K$, $1K$ and $10K$ are considered.
>
> We additionally emphasize that the results for Ornshtein-Uhlenbeck (our Figure 3 vs. [21, Figure 3]) are similar: the same "peak" at $D=3$ appears.
>
> **(5) Large variance.**
>
> We suppose that the variance for our method is higher than for particle simulation due to the fact our procedure is more influenced by randomness (random initialization of neural networks, stochastic gradient descent optimization). Note that the same applies to the the [Dual JKO] method by [21]: its variance is also higher than for particle simulation; see our Figure 3 or [21, Figure 3a].
>
> **(6) Accuracy (Table 1).**
>
> The reported accuracy and log-likelihood follows the conventional Bayesian logistic regression setup [38,43]. We have train and test datasets $S_{train}, S_{test} = ((x^{train}_i, y^{train}_i)), ((x^{test}_i, y^{test}_i))$ for binary classification.
>
> Given the posterior distribution $\theta \sim p(\theta|S_{train})$, where $\theta$ is the set of parameters of a logistic model, we estimate how well the predictive distribution $p(S_{test}| \theta)$ fits the dataset $S_{test}$. To do so we estimate the predictive distributions for test dataset elements: $p((x_i^{test}, y_i^{test}) | \theta) = \text{logisticrule}((x_i^{test}, y_i^{test}), \theta)$ by sampling $\theta \sim p(\theta|S_{train})$ and averaging the computed logits. Next, we compute the appropriate log-likelihood and accuracy (fraction of predicted probabilities larger than $0.5$). The larger the log-likelihood (and accuracy), the better the learned Bayesian logistic regression model $p(\theta|S_{train})$ fits the test dataset $S_{test}$.
>
> **Concluding remarks.** Please respond to our post to let us know if the clarifications above suitably address your concerns about our work.  We are happy to address any remaining points during the discussion phase; if the responses above are sufficient, we kindly ask that you consider raising your score.

---

> > ### Author Response · Authors · 2021-08-23
> > **Comparison with the proximal recursion method by Caluya et al. (2018)**
> >
> > Dear reviewer,
> >
> > Following your suggestion, we added the proximal recursion method [EM PR] by Caluya et al. (2018) to our evaluation. We implemented the method on Python based on the publicly available [matlab code](https://github.com/kcaluya/UncertaintyPropagation).
> >
> > To qualitatively check the correctness of our implementation, we plotted the stationary distribution estimated by [EM PR] in the setting of section 4.1 for dimension $D=2$. In the [picture](https://drive.google.com/file/d/1gE0dmJsPV2u7vXy12LUprltNOCc9TqJ0/view?usp=sharing), the scattered points and the associated colorbar are obtained by [EM PR]; the contours correspond to ground truth distribution. The stationary distribution looks like the mixture of Gaussians, as expected.
> >
> > The quantitative results in the setting of section 4.2 for dimensions $D=1,\dots,8$ are presented in the tables below (lower SymKL is better). As in Subsection 4.2, we report the SymKL between the obtained and the true Ornstein-Uhlenbeck process distribution at $t = 0.5$ and $t = 0.9$. We repeat experiments 15 times. For [EM PR] we use number of particles $N = 400, 1000, 10000$. Larger particle sizes are computationally infeasible since the complexity of the method is proportional to $N^{2}$. Note that in the original paper, the authors considered $N=400$ and dimensions $D=1,2$.
> >
> > **(1) Results for** $t=0.5$
> >
> > | Method  | $D=1$ | $D=2$ | $D=3$ |$D=4$ | $D=5$ | $D=6$ | $D=7$ | $D=8$  |
> > |---|---|---|---|---|---|---|---|---|
> > [EM] 1K | -1.02 | -1.21 | -0.74 | -0.41 | -0.13 | 0.08 | 0.25 | 0.38 |
> > [EM] 10K | -2.72 | -1.78 | -1.17 | -0.73 | -0.4 | -0.14 | 0.06 | 0.27  |
> > [EM PR] 400 | -1.62 | -1.02 | -0.45 | -0.13 | -0.01 | 0.1 | 0.34 | 0.65 |
> > [EM PR] 1K | -2.05 | -1.26 | -0.72 | -0.24 | -0.03 | 0.05 | 0.17 | 0.51 |
> > [EM PR] 10K | -2.8 | -1.97 | -1.3 | -0.78 | -0.38 | -0.11 | 0.1 | 0.18 |
> > [Ours] | -3.13 | -2.58 | -2.29 | -2.09 | -2 | -1.75 | -1.85 | -1.83 |
> >
> > **(2) Results for** $t=0.9$
> >
> > | Method  | $D=1$ | $D=2$ | $D=3$ |$D=4$ | $D=5$ | $D=6$ | $D=7$ | $D=8$  |
> > |---|---|---|---|---|---|---|---|---|
> > [EM] 1K | -2 | -1.22 | -0.75 | -0.39 | -0.13 | 0.08 | 0.25 | 0.38 |
> > [EM] 10K | -2.76 | -1.78 | -1.17 | -0.74 | -0.4 | -0.14 | 0.06 | 0.23  |
> > [EM PR] 400 | -1.75 | -1 | -0.46 | -0.14 | -0.03 | 0.11 | 0.28 | 0.46 |
> > [EM PR] 1K | -2.15 | -1.26 | -0.71 | -0.26 | -0.02 | 0.11 | 0.18 | 0.32 |
> > [EM PR] 10K | -2.8 | -1.95 | -1.3 | -0.76 | -0.36 | -0.11 | 0.12 | 0.22 |
> > [Ours] | -3.24 | -2.39 | -2.36 | -2.08 | -2.02 | -2.11 | -2.07 | -1.99 |
> >
> >  For better perception, we also add the links to visualizations of results (analogously to Figure 3 of our paper):
> >
> > (1) Comparison at $t=0.5$ : [link](https://drive.google.com/file/d/1Q0rVvh9eojd_E_9qDDZmbQvUM2aYLhUK/view?usp=sharing),
> >
> > (2) Comparison at $t = 0.9$ : [link](https://drive.google.com/file/d/1UbvIbZ2a7vQsNt2Owm8KoDhXb3gB_FpW/view?usp=sharing).
> >
> > The evaluation shows that [EM PR] technique performs comparably to vanilla [EM] and is outperformed by our method. Please respond to our post to let us know if the experiment above suitably addresses your concerns about our work. We are happy to address any remaining points during the discussion phase; if the responses above are sufficient, we kindly ask that you consider raising your score.

---

> > > ### Comment · Reviewer_tTtz · 2021-08-25
> > > **thank you**
> > >
> > > Thanks for the detailed response. I still think the scope of this paper is restricted. I think my score is fair and I will keep it unchanged.

---

### Official Review · Reviewer_FY52 · 2021-07-17

**Rating:** 5
**Confidence:** 4

**Summary:**

The Fokker-Planck equation can be treated as gradient descent over entropy functionals in Wasserstein space. To solve this problem, Jordan Kinderlehrer and Otto proposed the JKO scheme. Later, through the connection between the Brenier theorem and this scheme, [10] treat it as an optimization problem which aims at finding the convex Brenier functional $\psi$ in equation (7). However, the solution given by [10] is usually intractable in high dimensional space. Thus, this paper proposes to use the input convex neural networks (ICNNs) proposed by [5] to replace $\psi$. And then use SGD to optimize the model parameter. This forms the core contribution of this work. Additionally, the experiments are interesting.

**Main Review:**

- My main concern about this paper is its lack of novelty. As summarized above, the core contribution is to replace the convex functional $\psi$ in equation (7) proposed by [10] with the input convex neural networks (ICNNs) proposed by [5]. It is a common trick to replace the convex Brenier potential with the ICNN network in optimal transport community. The only difference is the background of the applications.
- In theorem 1, how to compute the high dimensional determinant $det \nabla T(x_n)$?
- In Sec. 4.1, since $W_2(\rho_{k-1}, \rho_k)$ for different $k$ has been computed during the optimization, it should be reported. The plot can help illustrate the performance of the proposed method.
- The comparison between the proposed method and [10] in low dimensional space in terms of memory usage/training time/accuray is necessary, because it is a direct generalization of [10].

**Time Spent Reviewing:**

8

---

> ### Author Response · Authors · 2021-08-07
> **Answers to Reviewer FY52**
>
> Dear reviewer,
>
> Thanks for your thoughtful review. Please find responses to your comments and questions below.
>
> **(1) Novelty.**
>
> The main novelty of our method lies in the idea of parameterizing convex potentials using ICNNs, along with addressing the practical challenges in the setting of JKO scheme that come with such new parameterization (Section 3.1-3.3).
>
> In fact, the crux of contributions in [10] is also in deriving a new but discrete and less practical parameterization for the convex potentials, and the idea of parameterizing measures as pushforwards of another measure has been well-known [42].
> To further illustrate their method's difference from ours, in [10], discrete analogues need to be defined for the Monge-Ampere operator ${\varphi \mapsto \det(\Delta\varphi)}$ as well as the pushforward map $\nabla \varphi_\sharp$ since they parameterize $\varphi$ as real-valued functions supported on a finite set of points.
> In comparison, thanks to ICNNs and auto-differentiation, we can compute $\det(\Delta\varphi)$ and $\nabla \varphi_\sharp$ exactly without introducing additional levels of discretization errors. We also do not need to assume measures supported on a discrete spatial domain which is a bottleneck for most existing methods, including [10], to scale to higher dimensions. As we demonstrate in Section 4, our method scales with dimensions much better than the existing approaches for computing gradient flows.
>
> **(2) Computational cost of $\log\det\nabla T$.**
>
> In all our experiments, we compute $\log\det\nabla T$ exactly, see lines 261-262 and Table 2 for discussion of the computational complexity.
> We noted in lines 287-288 on ideas using stochastic approximation to speed up the computation.
>
> **(3) Plots of** $W_{2}^{2}(\rho_{k},\rho_{k+1})$
>
> In the discretized Wasserstein gradient flow the value $W_{2}^{2}(\rho_k,\rho_{k+1})$ might be viewed as the analog to the gradient norm  $\|x_k - x_{k+1}\|^2_2$ in the usual Euclidean gradient descent $x_{k+1}\leftarrow x_{k}-\tau \nabla f(x)$. In our experiments, we observed a similar behaviour: $W_{2}^{2}(\rho_k,\rho_{k+1})$ decreases to zero as $k$ increases, similar to its Euclidean counterpart.
>
> For completeness, we provide an example of the evolution of $W_{2}^{2}(\rho_k,\rho_{k+1})$ for the convergence to stationary distribution experiment (Section 4.1) in $D=13$:
>
> $[1.51, 6.58\cdot 10^{-1}, 2.34\cdot 10^{-1}, 8.36\cdot 10^{-2}, 2.92\cdot 10^{-2}, 1.13\cdot 10^{-2}, 6.66\cdot 10^{-3}, 4.35\cdot 10^{-3}, 1.83\cdot 10^{-3}, 1.38\cdot 10^{-3}].$
>
> For brevity we report values only for steps $k=1,6,11,\dots,46$. Results are averaged over 6 random experiments. We can add this experiment to the main text in the final version if needed.
>
> **(4) Comparison with [10] in memory/time/metrics.**
>
> We see no need of adding [10] to the comparison, since, as we mentioned above, they require spatial discretization that cannot scale to moderate dimensions. In our view, for small dimensions, existing methods for domain discretization or particle simulation might run faster (see lines 271-273) with accuracy comparable to ours (see [EM 50K] method in Figure 3a or performance of [BBF] in non-linear filtering Figure 4a). However, in higher dimensions, which are the main target of our paper, these discretizations are computationally infeasible to achieve good accuracy, as we demonstrate in Section 4.1, 4.2.
>
> **Concluding remarks.** Please respond to our post to let us know if the clarifications above suitably address your concerns about our work.  We are happy to address any remaining points during the discussion phase; if the responses above are sufficient, we kindly ask that you consider raising your score.

---

> > ### Comment · Reviewer_FY52 · 2021-08-28
> > **thanks for the response**
> >
> > I thank the authors for the detailed responses. I still think the idea of using ICNN to replace the convex function $\psi$ is not so fresh, also I agree with the other reviewers that the scope of this paper is restricted. Thus, I decide to keep my score unchanged.

---

> > > ### Author Response · Authors · 2021-08-31
> > > **Response**
> > >
> > > While the general idea of using ICNN parametrization for convex function is not fresh, we argue that our particular idea to apply ICNN to gradient flows is indeed novel and provides a notable improvement over existing methods, especially in high dimensions.
> > >
> > > Following your suggestion, we planned to include comparison with [10], a related approach that uses spatial discretization instead of ICNNs - see a detailed discussion on such discretization compared to ours in the first point of our original response (https://openreview.net/forum?id=nlLjIuHsMHp&noteId=mCxKKRtd4KF).
> > > There is no publicly available code of [10], so we tried to implement the method ourselves. Unfortunately, it turned out that [10] lacks several important technical details for practical implementation. For example, the minimization (see equation (4.41) in [10]) with respect discrete pushforward operator $G_{\phi\sharp\mu}$ (see definition (3.28) in [10]) requires optimization with respect to all valid gradient maps $G_{\varphi}(p)$ (see def. 2.5), but the exact procedure for this is not explained in the paper.
> > >
> > > We emphasize that the method of [10] was tested only in 2D and the evolving measures should be supported on a convex polygon (see assumption G1 in [10]). It is not clear whether [10] can be implemented for higher dimensions since handling convex polytopes is very expensive for higher dimensions. In our paper, we aim at the high dimensional setting where distributions can be supported on the entire space. Therefore, we argue that comparison with [10] is not necessary.

---

### Official Review · Reviewer_JdcD · 2021-07-19

**Rating:** 6
**Confidence:** 4

**Summary:**

In this work the authors introduce an innovative discretisation of the Jordan Kinderleher Otto scheme (which describes the evolution of the gradient flow of relative entropy in the Wasserstein metric).  To do so the authors iteratively construct a transformation map as a composition of gradients of input-convex neural networks, which satisfy a universal approximation theory among convex functions.  Using Brenier's theorem, an iterative variational formulation for the density at discrete time-steps is obtained, from which one can generate samples and or evaluate an unnormalised density.

A variety of applications are presented, including finding the stationary distribution, as well as nonlinear filtering.

**Limitations And Societal Impact:**

this has been adequately addressed

**Main Review:**

This submission brings two important ideas together -- the idea of using input-convex neural networks with Brenier's theorem to formulate a variational problem for the density.   While straightforward, it is a nice idea which provides a means of evaluating the density of an overdamped Langevin dynamics which overcomes the curse of dimensionality associated with spatial discretisation approaches.

While I find the idea interesting and the submission is clearly written, I would argue that it provides a very elegant solution to a quite narrow problem.   Evaluating the density of an SDE beyond the usual overdamped langevin SDE is not supported (granted one can adjust the metric tensor to accomodate some cases), in particular, non-ergodic diffusions cannot be approached via this formulation.

Secondly, for problems which involve stationary sampling as presented in the examples, we are severely limited by the various costs, mainly the quadratic cost in time-step which forbids long time -simulations.

Finally, it's worth remembering that this is still a discretisation in time, and it is not clear from this manuscript whether discretisation error induced by finite h has really been considered or studied.  I feel that this should have been addressed more carefully in the examples.

**Time Spent Reviewing:**

2

---

> ### Author Response · Authors · 2021-08-07
> **Answers to Reviewer JdcD**
>
> Dear reviewer,
>
> Thanks for your thoughtful review. Please find responses to your comments and questions below.
>
> **(1) Extensions to other diffusions.**
>
> While non-ergodic diffusions probably cannot be approached via our formulation, we emphasize that $\mathcal{F}_{\text{FP}}$ itself already covers several real-world, practical instances of diffusion as we have shown.
>
> In addition, in lines 153-155 we note that our method extends to other functionals $\mathcal{F}(\rho)$ that admit stochastic estimation from random batches from $\rho$.
>
> Interaction energy ${\mathcal{I}(\rho)=\int W(x-y)d\rho(x)d\rho(y)}$ is a possible example of such a functional appearing in many practical tasks. Similar to potential energy $\mathcal{U}(\rho)$ (line 95), it admits estimates from batches $x_i\sim\rho$ and $y_j\sim\rho$, i.e. $\mathcal{I}(\rho)\approx \frac{1}{N^{2}}\sum_{i=1}^{N}\sum_{j=1}^{N}W(x_{i}-y_{j})$.
>
>
> The list of suitable functionals $\mathcal{F}(\rho)$ is much wider, see [10, eq. (1.9-1.11)] for more examples. Studying applications of our method to these functionals is a promising avenue for future work.
>
> **(2) Long time-simulations.**
>
> We agree that our method might not work fast for long time simulations due to stacking multiple networks. To tackle this issue, we described possible improvements such as approximating the chain of pushforward maps with an invertible network (lines 284-286).
>
> **(3) Discretization error induced by finite h.**
>
> Following your request, we conducted the following experiment. In the setting of Ornstein-Uhlenbeck Process (Section 4.2) in $D=13$ we launched our method with varying time steps $h\in[0.02, 0.04, 0.05, 0.08, 0.1, 0.2, 0.4]$. We computed log of SymKL at $t=0.4$ between the computed and ground truth distribution. The obtained values (average over 6 runs) are
> $[0.016, 0.029, 0.058, 0.099, 0.3, 0.89]$. The trend of the results is clear and matches those of Euclidean gradient descent: the lower the step size is (learning rate), the better the time-discretized flow matches the true continuous flow. We will add these results in the final version of the paper.
>
> **Concluding remarks.** Please respond to our post to let us know if the clarifications above suitably address your concerns about our work.  We are happy to address any remaining points during the discussion phase; if the responses above are sufficient, we kindly ask that you consider raising your score.

---

> > ### Comment · Reviewer_JdcD · 2021-08-23
> > **Answer to Authors**
> >
> > Thanks for your detailed replies and for taking into account my comments, particularly for the discretisation error induced by finite $h$ which is quite interesting to see.
> >
> > In light of the my comments, the reviewers comments and your replies, I maintain that this is interesting work of some significance, but remains of somewhat narrow scope.   Based on this I shall leave my score as is.

---

### Author Response · Authors · 2021-09-11
**Scope of our work**

Dear area chair and reviewers,

Before discussion closes, we wanted to share some discussion and evidence involving the broad applicability of our work; we do not find that it is too narrow.

Overall, we provide an efficient method to model diffusion processes arising in many practical tasks. We considered applications to popular Bayesian tasks, namely **unnormalized posterior sampling** (section 4.3), and **nonlinear filtering** (section 4.4). Extending to other species of gradient flow might be interesting mathematically, but *already these applications form a representative set* with potential for practical impact in machine learning.

Below we detail several other potential applications of the machinery we propose.

**Population dynamics.** In this task, one needs to recover the potential energy $\Phi(x)$ included in the Fokker-Planck free energy functional $\mathcal{F}_{\text{FP}}(\rho)$ (see (5) in our paper) based on samples from the diffusion obtained at timesteps $t_1, t_2, \dots t_n$ (see [1] for reference). This setting can be found in computational biology (see section 6.3 of [1]). Recently, a paper [2] appeared that uses the idea of utilizing ICNN-powered JKO to model population dynamics.

**Reinforcement learning.** Wasserstein gradient flows provide a theoretically-grounded way to optimize an agent policy in reinforcement learning, see [3,4]. The idea of the method is to maximize the expected total reward (see (10) in [3]) using the gradient flow associated with the Fokker-Planck functional (see (12) in [3]). The authors of the original paper proposed discrete particle approximation method to solve the underlying JKO scheme. Substituting their approach with our ICNN-based JKO can potentially improve the method.

**Refining Generative Adversarial Networks.** In the GAN setting, given trained generator $G$ and discriminator $D$, one can improve the samples from $G$ by $D$ via considering a gradient flow w.r.t. entropy-regularized $f$-divergence between real and generated data distribution (see [5], in particular, formula (4) for reference). Using KL-divergence makes the gradient flow consistent with our method: the functional $\mathcal{F}$ defining the flow has only entropy and potential energy terms. The usage of our method instead of particle simulation may improve the generator model.

Taking all these applications into account, we disagree that the scope of our paper is narrow. We kindly ask you to take this into account when making the final decision.

[1] Hashimoto et. al. Learning
Population-Level Diffusions with Generative Recurrent
Networks. http://proceedings.mlr.press/v48/hashimoto16.pdf

[2] Bunne et. al. JKOnet: Proximal Optimal Transport Modeling of Population Dynamics. https://arxiv.org/pdf/2106.06345.pdf

[3] Zhang et. al. Policy Optimization as Wasserstein Gradient Flows. https://arxiv.org/pdf/1808.03030.pdf

[4] Richemond, P. H., \& Maginnis, B. (2017). On wasserstein reinforcement learning and the fokker-planck equation. https://arxiv.org/pdf/1712.07185.pdf

[5] Ansari et. al. Refining Deep Generative Models via Discriminator Gradient Flow. https://arxiv.org/pdf/2012.00780.pdf

---

### Decision · Program_Chairs · 2021-09-27

**Decision:**

Accept (Poster)

**Comment:**

This work focuses on a new method to approximate Wasserstein gradient flows, allowing to sample from and compute the density at each time step. The use of ICNN to play the role of the convex potential allows to discretize the time dimension efficiently. Based on the overall evaluation by the reviewers - which is mostly positive - and my own, I recommend to accept this work.

There was a serious and fruitful discussion between reviewers and authors, the latter providing interesting perspective on the motivation of their work, that should be included in a camera-ready version.